# Culture-Independent PCR Detection and Differentiation of *Mycobacteria* spp. in Antemortem Respiratory Samples from African Elephants (*Loxodonta Africana*) and Rhinoceros (*Ceratotherium Simum*, *Diceros Bicornis*) in South Africa

**DOI:** 10.3390/pathogens11060709

**Published:** 2022-06-20

**Authors:** Wynand J. Goosen, Charlene Clarke, Léanie Kleynhans, Tanya J. Kerr, Peter Buss, Michele A. Miller

**Affiliations:** 1DSI-NRF Centre of Excellence for Biomedical Tuberculosis Research, South African Medical Research Council Centre for Tuberculosis Research, Division of Molecular Biology and Human Genetics, Faculty of Medicine and Health Sciences, Stellenbosch University, P.O. Box 241, Cape Town 8000, South Africa; cclarke@sun.ac.za (C.C.); leaniek@sun.ac.za (L.K.); tjkerr@sun.ac.za (T.J.K.); miller@sun.ac.za (M.A.M.); 2Veterinary Wildlife Services, Kruger National Park, South African National Parks, Skukuza 1350, South Africa; peter.buss@sanparks.org

**Keywords:** African elephants, bronchioloalveolar lavage, GeneXpert MTB/RIF Ultra, Hain CM*direct* V1.0 LPA, *ku* PCR, *mycobacterium tuberculosis* complex, non-tuberculous mycobacteria, rhinoceros, *rpoB* PCR, trunk wash

## Abstract

Since certain *Mycobacterium tuberculosis* complex (MTBC) members, such as *M. bovis*, are endemic in specific South African wildlife reserves and zoos, cases of clinically important nontuberculous mycobacteria (NTM) in wildlife may be neglected. Additionally, due to the inability of tests to differentiate between the host responses to MTBC and NTM, the diagnosis of MTBC may be confounded by the presence of NTMs. This may hinder control efforts. These constraints highlight the need for enhanced rapid detection and differentiation methods for MTBC and NTM, especially in high MTBC burden areas. We evaluated the use of the GeneXpert MTB/RIF Ultra, the Hain CM*direct* V1.0 line probe assay, and novel amplicon sequencing PCRs targeting the mycobacterial *rpoB* and *ku* gene targets, directly on antemortem African elephant (*n* = 26) bronchoalveolar lavage fluid (BALF) (*n* = 22) and trunk washes (*n* = 21) and rhinoceros (*n* = 23) BALF (*n* = 23), with known MTBC culture-positive and NTM culture-positive results. Our findings suggest that the Ultra is the most sensitive diagnostic test for MTBC DNA detection directly in raw antemortem respiratory specimens and that the *rpoB* PCR is ideal for *Mycobacterium* genus DNA detection and species identification through amplicon sequencing.

## 1. Introduction

Mycobacteria are a diverse group of microorganisms that can be found in practically every environmental niche [1]. They are divided into two categories: *Mycobacterium tuberculosis* complex (MTBC) and non-tuberculous mycobacteria (NTM) [2]. *Mycobacterium tuberculosis*, *M. bovis*, *M. orygis*, *M. bovis* bacillus Calmette–Guerin (BCG), *M. africanum*, *M. cannettii*, *M. pinnipedii*, *M. caprae*, and *M. microti* are all members of the MTBC. *Mycobacterium bovis* and *M. tuberculosis* are well known for causing chronic infectious disease in people and animals, including livestock and wildlife species [3,4,5,6]. Cases of *M. bovis* in different South African (SA) wildlife, including African buffaloes, African lions, African wild dogs, and rhinoceros, are sporadically reported. The disease control efforts for buffaloes include test-and-slaughter strategies, and quarantine for endangered species [3,5,7,8,9]. Many of the high TB burden countries are heavily dependent on animal-related industries, such as tourism and agriculture, leading to habitat encroachment and increased opportunities for disease transmission at the animal–human interface. This phenomenon has recently been highlighted by the unexpected discovery of a fatal *M. tuberculosis* infection in an African elephant (*Loxodonta africana*) in Kruger National Park (KNP), which is endemic for *M. bovis* [10,11].

Non-tuberculous mycobacteria, also known as environmental mycobacteria, are natural inhabitants of the environment and comprise more than 150 species listed on public bacterial databases (http://www.bacterio.net (accessed on 19 May 2022)). Interestingly, more than a third of these species have been implicated in diseases in livestock, wildlife, and humans [12]. Numerous animal and human NTM infections are reported globally, probably due to their ubiquitous presence and opportunistic nature [13,14]. The *Mycobacterium avium* complex (MAC) organisms, consisting of *M. avium* subsp. *avium*, *M. avium* subsp. *paratuberculosis*, *M. avium* subsp. *hominissuis*, *M. intracellulare*, *M. sylvaticum*, *M. colombiense*, *M. bouchedurhonense*, *M. timonense*, *M. chimaera*, *M. arosiense*, *M. yongonense*, and *M. marseillense,* are the most well-known opportunistic NTMs reported in animals and humans [15]. In addition, *M. kansasii*, *M. marinum*, and *M. ulcerans* commonly cause opportunistic NTM disease. Infections with *M. kansasii* can cause pathological signs resembling the *M. bovis*-associated disease in animals [12,16]. Another NTM, *M. scrofulaceum* has been found in the lymph nodes of cattle, buffaloes, farmed deer, swine, wild pigs, patas monkeys, fish, and mice [17]. These organisms can spread indirectly through contaminated environments, as is evident by the isolation of mycobacteria from cattle and mice faeces [17]. 

In Africa, most of the NTMs are reported as incidental findings in livestock with gross pathological lesions identified in slaughterhouses, when tissues are submitted for *M. bovis* surveillance [18,19]. There are substantially fewer reports of NTMs isolated from free-ranging wildlife. Therefore, the extent and distribution of NTM infections among the African wildlife species are largely unknown. In SA, the *M. avium* subspecies *paratuberculosis*, *M. terrae*, *M. nonchromogenicum*, *M. vaccae*/*M. vanbaalenii*, and unidentified species closely related to *M. moriokense,* have been isolated from African buffaloes, livestock, and their environments, indicating that NTMs may be exchanged at environment–animal interfaces [20]. Besides being opportunistic pathogens, some of the NTM species may colonize the host without development of disease, but instead priming the host’s immune system, confounding the immuno-diagnosis of bovine tuberculosis (bTB) due to cross-reactivity to shared antigens [21,22]. The genetic investigations into NTM isolates from the samples collected from various wildlife species showed the presence of orthologous genes, such as *ESAT-6* and *CFP-10,* situated in the *ESX-1* to *ESX-5* regions [2]. The occurrence of positive *M. bovis* immunological test results in these animals suggest that NTMs potentially interfere with bTB diagnostic assays. Since *M. bovis* is endemic in several of the SA wildlife reserves and zoos, the cases of NTM infection in wildlife may be neglected, due to the primary focus being on *M. bovis’* diagnosis. However, due to the inability of tests to differentiate the host responses towards MTBC and NTMs, due to the interference of NTMs, this may hinder control efforts. Unfortunately, correctly identifying a NTM requires mycobacterial culture with a minimum 6–8-week incubation period and further genetic speciation [23]. Previously, using elephants’ and rhinoceros’ respiratory samples, mycobacterial culture was shown to have a lower limit of detection of 1000 colony forming units (CFU) for culturing *M. bovis,* and 100 CFU for *M. tuberculosis* [23,24]. This constraint highlights the need for enhanced and rapid MTBC and NTM detection and differentiation assays, especially in the high *M. bovis* burden areas.

Assays using PCR have been used to rapidly detect and differentiate between MTBC and NTM. Recently, Cepheid’s GeneXpert MTB/RIF Ultra assay (Ultra) has been shown to provide rapid detection of MTBC DNA in the tissue and respiratory samples collected from infected African buffaloes, African elephants, and rhinoceros [5,6]. The Hain GenoType CM*direct* VER 1.0 line probe assay (Hain LPA) can also lead to sensitive and specific detection of the DNA from the *Mycobacterium* genus organisms, MTBC (without differentiation), and the differentiation of more than 20 clinically relevant NTMs directly from human patient specimens [25]. Although *16S* rRNA sequencing of *Mycobacterium* cultures has been used for speciation, there are increasing reports that this technique is sub-optimal [26]. More recent studies, using hundreds of strains of M*ycobacterium* spp. isolates, have demonstrated that the combined PCR amplification of highly conserved regions of the *Mycobacterium ku* and *rpoB* genes, using primers specifically designed for the *Mycobacterium* genus, have superior performance compared to the *16S* rRNA amplicon sequencing for genus detection and the speciation of low-level mixed microbial populations [26,27,28,29]. Consequently, these findings have major implications for the field of targeted, next generation sequencing directly from clinical samples, especially for the improved surveillance of clinically important *Mycobacteria* spp. Therefore, the aims of this pilot study were to evaluate the use of (1) the Ultra and Hain LPA for rapid MTBC DNA detection; (2) the Hain LPA for NTM DNA detection and species differentiation without required sequencing; and (3) *ku* and *rpoB* amplicon sequencing for the *Mycobacterium* genus DNA detection and species differentiation, directly from antemortem respiratory samples. These samples were collected from known MTBC and NTM culture-confirmed positive African elephants and rhinoceros from KNP and a zoo in SA.

## 2. Results 

### 2.1. Mycobacterial Culture Results 

The respiratory samples from six animals (five elephants and one white rhinoceros) were defined as the culture-confirmed MTBC positive cohort for this study (Figure 1 and Table 1). Six of the eight elephant respiratory samples, three bronchoalveolar lavage fluid (BALF) samples (including one duplicate sample), and three trunk washes from five MTBC-infected elephants (free-ranging and zoo), were confirmed to contain viable MTBC through mycobacterial culture and speciation. The remaining two respiratory samples (TW and BALF) did not grow MTBC, but rather *M. elephantis* and *M. stomatepiae* were isolated (Table 1). The infected animals included: (1) two zoo elephants (18/85 and 18/177), from which *M. tuberculosis* was cultured from the first animal’s (18/85) BALF sample, *M. tuberculosis* and *M. africanum* from the second elephant’s (18/177) duplicate BALF samples and *M. elephantis* from its TW sample, respectively; and (2) three free-ranging KNP elephants (18/527, 18/533, and 18/538), from which *M. bovis* was cultured from each animal’s TW sample (Table 1). Even though the KNP elephant 18/527 TW sample contained viable culturable *M. bovis*, its BALF sample did not (Table 1). Lastly, *M. bovis* was also cultured from a BALF sample collected from a white rhinoceros (19/46) from KNP (Table 1). 

Respiratory samples from nine animals (eight elephants and one white rhinoceros) were defined as the culture-confirmed NTM positive cohort for this study (Figure 1 and Table 1). Ten respiratory samples (seven TW and three BALF) from eight elephants and a single (BALF) sample from one rhinoceros were confirmed to contain viable NTMs through mycobacterial culture (Table 1). The *Mycobacterium mantenii* was successfully isolated and characterized from a KNP elephant’s (19/460) BALF sample and the *M. abscessus* strain from its TW sample. Similarly, *M. interjectum* was isolated and characterized from a KNP elephant’s (18/530) BALF sample and the *M. avium* complex from its TW sample (Table 1). Non-tuberculous mycobacteria isolated from the TW samples collected from the remaining four KNP elephants and one zoo elephant included the *M. avium* complex strain (18/173), *M. mageritense* strain (18/532), *M. intracellulare* (18/534 and 18/539), and *M. fortuitum* strain (21/496). The elephant (18/176) and rhinoceros (18/31) BALF samples were culture positive for the *M. foliorum* and *M. scrofulaceum* strain, respectively (Table 1). 

Additionally, 25 respiratory samples (15 BALF and 10 TW) from 13 elephants and 21 BALF samples from 21 rhinoceros (20 white- and 1 black rhinoceros) were considered mycobacterial culture negative by the BACTEC 960 MGIT system and defined as the culture negative cohort in this study (Figure 1; Appendix A).

### 2.2. Presence of ESAT-6/CFP-10 in All Mycobacterial Cultures

All of the MTBC culture isolates from seven respiratory samples (one rhinoceros and five elephants, including one duplicate BALF sample), were PCR-positive for *ESAT-6* and *CFP-10* (Table 1). Nine of the thirteen NTMs isolated by the respiratory sample culture were also PCR-positive for *ESAT-6* and *CFP-10* (one rhinoceros and ten African elephants) (Table 1). However, one of the PCR-positive NTM strains (*M. elephantis*) was cultured from a TW sample from an elephant (18/177), with *M. tuberculosis* and *M. africanum* isolated by culture in the BALF samples (Table 1). 

Within the culture negative cohort, the presence of *ESAT-6* and *CFP-10* was detected in 7 out of the 46 respiratory sample cultures (13 elephants and 21 rhinoceros; Appendix A). For these *ESAT-6/CFP-10* positive samples, the potential presence of the following *Mycobacterium* species was also identified by PCR amplicon sequencing directly from the respiratory samples: (1) *M. africanum*; (2) *M. bovis*; (3) *M. interjectum*; (4) *M. intracellulare;* (5) *M. avium* complex; and (6) *M. orygis* (Appendix A). The target amplification was confirmed through Sanger sequencing and using the NCBI’s Basic Local Alignment Search Tool for nucleotide (BLASTn) [30].

### 2.3. Nucleic Acid Amplification Test Results on Raw Respiratory Samples

#### 2.3.1. Ultra and Hain LPA for MTBC DNA Detection

The Ultra successfully identified all five of the infected elephants and the one infected rhinoceros as MTBC infected, which included four positive BALF and three elephant TW samples out of the nine respiratory samples (7/9) (Table 1). The Hain LPA also correctly identified three of the MTBC-infected elephants (18/85, 18/527, and 18/533), based on one positive TW sample and two positive BALF samples. The *M. bovis*-positive rhinoceros (19/46) was also identified as MTBC positive by the Hain LPA, using its BALF sample (Table 1). Three of the four positive MTBC results on the Hain LPA agreed with the Ultra results, with a discordant result (negative Ultra; MTBC and/or *M. fortuitum* group identified by the Hain LPA) for the BALF sample of the *M. bovis*-positive elephant (18/527). This BALF sample was *ESAT-6/CFP-10* negative and *M. stomatepiae* was isolated (Table 1). The TW sample from the *M. tuberculosis*-infected zoo elephant (18/177), based on the BALF culture, only contained culturable *M. elephantis*, which was negative on the Ultra, Hain LPA, and *ESAT-6/CFP-10* PCRs (Table 1). 

When the Ultra and Hain LPA results were compared, using a two-tailed z-test, there was a significant difference (*p* < 0.00001) between these two tests for MTBC detection within the confirmed MTBC-infected cohort. However, agreement between the Ultra and Hain LPA for MTBC DNA detection from all of the specimens (regardless of culture outcome) was “substantial” (κ = 0.75, 95% CI 0.55–0.96: standard error (SE) = 0.10) [31]. Both the Ultra and Hain LPA identified the same three elephants (18/173, 19/460, and 18/534) as MTBC infected, based on one BALF and two TW samples, although the culture isolates categorized these individuals in the NTM positive cohort (Table 1). All three of these respiratory samples were also positive for *ESAT-6/CFP-10* by PCR. Similarly, within the culture negative cohort, the Ultra and Hain LPA detected the same four elephants as MTBC infected, based on the results from the three BALF samples and one TW sample (all *ESAT-6/CFP-10* positive) (Appendix A). 

A comparison between the Ultra and Hain LPA results with culture detection showed “substantial” agreement between the Ultra and culture for MTBC detection (κ = 0.61, 95% CI 0.36–0.86, SE = 0.13), although lower than agreement between the Ultra and Hain LPA results (κ = 0.75). However, there was only “slight” agreement between the Hain LPA and culture results (κ = 0.23, 95% CI-0.07–0.54: SE = 0.16). When the results of the Ultra and Hain LPA were combined and compared to culture for MTBC detection, there was only “moderate agreement” (κ = 0.58, 95% CI 0.33–0.83: SE = 0.13).

#### 2.3.2. Hain LPA NTM DNA Detection and Species Differentiation

The Hain LPA identified all eleven of the respiratory samples from the culture-confirmed NTM positive cohort (eight elephants and one rhinoceros), as containing NTM DNA (11/11). Eight of the samples (8/11) were also positive for *ESAT-6/CFP-10* amplification and three of those included samples from the elephants (18/173, 19/460, and 18/534) that were also identified as containing MTBC DNA by the Ultra and Hain LPA (Table 1). The Hain LPA detected mixed NTM DNA (>1 NTM strain) within the samples from these three KNP elephants, as well as an additional KNP elephant (18/530) and KNP rhinoceros (18/31) within the NTM positive cohort (Table 1). Similarly, within the MTBC-infected cohort, the Hain LPA also detected NTM DNA in eight out of nine respiratory samples, with five of the respiratory samples showing a mixture of NTM DNA (Table 1). Within the culture negative cohort, NTM DNA was detected by the Hain LPA in 19 of the 46 respiratory samples. These included four samples from the four elephants also identified as containing MTBC DNA by both the Ultra and Hain LPA. Three of these four elephant samples, and six additional elephant respiratory samples, were shown to have a mixture of NTM DNA based on the Hain LPA (Appendix A). When the Hain LPA was compared to the culture results for the detection of NTM using all of the respiratory samples, only “fair” agreement was observed (Table 2). However, when the identified NTM species by the Hain LPA were compared with culture results, the Hain LPA only correctly identified one culture isolate (KNP elephant 19/460, TW, *M. abscessus* strain) (Table 1).

#### 2.3.3. Ku and rpoB Amplicon Sequencing for *Mycobacterium* Genus Detection and Speciation

Nine of the respiratory samples, confirmed to contain MTBC (*M. tuberculosis, M. africanum, M. bovis*) by culture, showed an amplification in the *ku* and *rpoB* PCRs, which identified the presence of the *Mycobacterium* genus DNA (Table 1). Upon amplicon sequencing, both PCRs correctly identified the same MTBC species, directly from respiratory samples, as by culture (Table 1). Discordant culture/*ku* PCR results were observed for two of the elephant (18/177 and 18/527) TW and BALF samples. The elephant (18/177) TW was identified as containing *M. elephantis* (*ESAT-6/CFP-10* positive) by culture and *rpoB* PCR, but as the “*M. fortuitum* group” by the *ku* PCR (Table 1). Similarly, the elephant (18/527) BALF was identified as containing *M. stomatepiae* (*ESAT-6/CFP-10* negative) by culture and *rpoB* PCR, but “mixed NTMs/*M. smegmatis”* by *ku* PCR (Table 1). When comparing the results from the *ku* and *rpoB* PCRs, there was no significant difference (two-tailed z-test, *p* = 1) between these two tests for the *Mycobacterium* genus detection and the MTBC species identification within the culture-confirmed MTBC-infected cohort. 

Both the *ku* and *rpoB* PCRs identified the presence of the *Mycobacterium* genus DNA in all but one of the eleven respiratory samples from the NTM-positive cohort. The TW sample from the KNP elephant (18/532) was culture positive for the *M. mageritense* strain (*ESAT-6/CFP-10* negative) but identified as the *M. fortuitum* group by the Hain LPA (Table 1). The *rpoB* amplicon sequences predicted the same species of NTM as culture, but directly from the respiratory samples; the *ku* amplicon sequences identified five of the eleven NTM species assigned after culture (Table 1). Although there was no statistical difference (*p* = 1.0) between the two PCRs for *Mycobacterium* genus detection, there was a significant difference in the ability to identify NTMs (*p* < 0.02). 

Within the culture negative cohort, the presence of the *Mycobacterium* genus was detected by both PCRs in the same 15 elephant respiratory samples (also all were Hain LPA positive). Two of the Hain LPA positive elephant (18/537 TW and 18/539 BALF) samples were *rpoB* PCR positive, but *ku* PCR negative; in addition, one elephant 18/157 TW sample that was also Hain LPA positive was *ku* PCR positive but *rpoB* PCR negative (Appendix A). Three respiratory samples from different elephants (18/255, 18/534 and 18/536) were shown to have *M. africanum, M. bovis,* and *M. orygis* DNA, respectively, by both the *ku* and *rpoB* PCRs, although these samples were culture negative but all *ESAT-6/CFP-10* PCR positive. These three samples were also positive for MTBC DNA by both the Ultra and Hain LPA (Appendix A). 

To evaluate the best test or combination of tests, including the Hain LPA for *Mycobacterium* genus detection, agreement analysis was performed (Table 2). For the *Mycobacterium* genus detection, “fair” to “moderate” agreement were observed between the culture and individual tests (*rpoB*-, *ku* PCR, and Hain LPA) as well as for the combinations of these tests versus culture results (Table 2). “Almost perfect” agreement was reported between the individual tests (*rpoB*, *ku* PCRs and Hain LPA), as well as test combinations (Table 2).

For the mycobacterial species identification compared to culture, the Hain LPA only correctly identified a single NTM-positive sample (as previously mentioned), and it was incapable of differentiating between the MTBCs. Compared to the culture results, the *rpoB* and *ku* PCRs both correctly identified the same 6/7 MTBC culture positive samples. For all of the NTM positive samples, the *rpoB* PCR correctly identified 10/13 NTM culture positive samples, with 29/46 culture negative samples also negative on the *rpoB* PCR (Table 1; Appendix A). The *ku* PCR correctly identified 4/13 NTM culture positive samples, and 30/46 of the culture negative samples were also negative on the *ku* PCR (Table 1; Appendix A). The four NTMs identified by the *ku* PCR were also correctly identified by the *rpoB* PCR (Table 1). 

## 3. Discussion

This pilot study described the successful culture isolation and genetic speciation of *M. tuberculosis*, *M. africanum* and *M. bovis* from antemortem respiratory samples collected from zoo and free-ranging African elephants and rhinoceros. Our recent culture isolation success was largely due to the combined use of conventional MGIT culture with a novel modified version called MGIT-TiKa [23,24], as well as improved isolate speciation through the simultaneous detection and sequencing of three different genetic markers (*16S* rRNA, *rpoB,* and *hsp65*). In the zoo elephant (18/85), *M. tuberculosis* had previously been isolated from the lung and lymph node tissue samples [5], and in this study, *M. tuberculosis* was also isolated from its antemortem BALF sample (Table 1). Notably, the elephant 18/85 BALF samples were collected before it was euthanized for tissue sample collection, removing the possibility of BALF contamination at necropsy. Moreover, in the same zoo, the simultaneous culture isolation of both *M. tuberculosis* and *M. africanum* from duplicate BALF samples was found in a single contact elephant 18/177. This was the only MTBC co-infection discovered among the elephants and rhinoceros in this study. Since this elephant was in a zoo environment in SA with known MTBC infections, the results were not that surprising [5]. The effects of MTBC co-infections on disease development are still largely unknown, especially in zoo settings, but certainly warrant further investigations. The successful culture isolation of *M. bovis* from three free-ranging African elephants and one rhinoceros in a *M. bovis* endemic wildlife reserve, such as KNP, were not unexpected, but it was surprising that it was cultured from paucibacillary antemortem respiratory samples. These findings suggest that possible MTBC shedding by elephants and rhinoceros may occur in zoos and wildlife reserves, however, it must also be noted that the isolation of *M. bovis* from elephant trunk wash samples could also have been due to contamination from other infected hosts, such as African buffaloes, shedding into the environment [32].

The isolation of various NTMs by culture from antemortem respiratory samples was also expected (Table 1), especially with the novel culturing approach used. It is noteworthy that some clinically important NTMs were isolated, along with supportive PCR evidence for the presence of ESAT-6 and CFP-10 virulence factors. These included *M. abscessus*, *M. avium* complex strains, *M. interjectum*, *M. fortuitum,* and *M. scrofulaceum* (Table 1). However, one limitation was that all of these animals were from MTBC endemic areas and the *M. tuberculosis* and *M. bovis* may have been present in small amounts below the detection threshold of 100 CFU for *M. tuberculosis* culture and 1000 CFU for *M. bovis*, but detectable by PCR [5,24]. This could result in positive *ESAT-6/CFP-10* PCR, but culture negative results (Table 1). For many of these NTMs, opportunistic human and animal infections have been reported [2,33,34,35]. Two cases of atypical mycobacteriosis, caused by *M. szulgai,* have been reported in zoo elephants, as well as disease caused by *M. kansasii* infection in a bontebok herd, which were all positive for antibodies to ESAT-6/CFP-10 [16,36]. These cases highlight the diagnostic challenges around the detection of NTMs and differentiation from MTBC infections. The need to differentiate mycobacterial species is becoming increasingly recognized, despite the fact that many NTMs may not cause disease, but could prime the host’s immune system, subsequently impeding the accurate diagnosis of MTBC infections, especially when using virulence factor proteins as test antigens [37]. 

An important finding, within the culture negative cohort, was the PCR detection and identification of MTBC DNA (*M. bovis*, *M. africanum, M. orygis*) in antemortem samples (Appendix A). Without this result, any positive immunological assay results would have been classified as false-positive, rather than MTBC infected. These findings were supported by the *ESAT-6/CFP-10* positive PCR results, combined by the simultaneous species’ identifications by three separate tests, the *rpoB* PCR, *ku* PCR, and the Hain LPA. However, a significant limitation is the inability of the PCRs to differentiate between live and dead bacteria, this may account for a positive PCR/culture negative result, which would be important for evaluating the transmission risk. Therefore, it is likely that shedding in MTBC-infected elephants and rhinoceros may occur more frequently than reported, due to the paucibacillary nature of antemortem respiratory samples and the overall sub-optimal sensitivity of culture [38,39].

The predictive ability of three candidate PCRs (Ultra, *rpoB* PCR, *ku* PCR) and one LPA, used directly on antemortem respiratory samples from culture-defined (MTBC, NTM, and negative) animal cohorts, was assessed to identify the most sensitive technique (Table 1). For the MTBC DNA detection, the Ultra qPCR was significantly (*p* < 0.00001) more sensitive than the Hain LPA, with samples from the MTBC culture positive cohort. In addition, one advantage was that the Ultra used raw samples as input material whereas the Hain LPA required DNA extraction. Discordant results between the tests were reported for elephant 18/537 (*M. bovis* culture positive TW) where the Ultra detected MTBC DNA only in the TW sample, whereas the Hain LPA detected MTBC DNA in the BALF sample (from which *M. stomatepiae* was isolated by culture). These findings highlight the possibility that the MTBC may be present in samples from infected animals but may be unculturable due to their paucibacillary nature and sub-optimal sensitivity of culture [24]. A sample’s bacterial load may also be reduced during sampling, sample handling, and storage. It is also possible that the TW sample reflected the overall respiratory load of bacilli and the possibility of not sampling the infected site using BAL. This highlights the importance of testing multiple samples.

Within the NTM culture positive and negative cohorts, MTBC DNA was also simultaneously detected by both the Ultra and Hain LPA, with positive amplification of the virulence factors for those samples. Again, this may have been dead bacteria or samples with small unculturable amounts of MTBC bacilli. Agreement between the Ultra, Hain LPA, and their combined use versus the culture results for all of the samples, showed that the Ultra was the most sensitive test for MTBC DNA detection. Combining the results from the Ultra and Hain LPA for MTBC DNA detection did not improve the overall detection, supporting the individual use of the Ultra for MTBC DNA detection directly from raw antemortem respiratory samples, which may identify infected animals that are culture negative.

The Hain LPA, for NTM DNA detection and species identification, identified all of the samples from the NTM positive cohort as containing NTM DNA (Table 1). Additionally, it also detected NTM DNA in the MTBC positive and culture negative cohort samples, which would be important for identifying the co-infections. The Hain LPA produced fair agreement compared to culture for the NTM detection, although there was no agreement with the culture for species’ identification. This observation may be due to the use of hybridization technology by the Hain LPA producing unclear subjective results, especially when used directly on raw animal samples, or further evidence of the selective pressure introduced by the mycobacterial culture during sample processing [24]. Most of the samples were identified by Hain LPA as containing DNA from a mixture of NTM species, but the cultures were positive for only a single NTM species. This could indicate that the NTM that was successfully cultured most likely had outcompeted the rest during incubation, or that they were eradicated prior to inoculation during the decontamination process [24].

Through the PCR amplification and subsequent amplicon sequencing of the *rpoB* and *ku* gene targets, using *Mycobacterium* specific primers, both of the PCRs correctly detected and identified all of the MTBC species directly from the antemortem respiratory samples within the culture-confirmed MTBC positive cohort. No significant difference (*p* = 1) for genus detection or species identification was reported between these PCRs within the MTBC positive cohort. Similarly, both PCRs detected the *Mycobacterium* genus DNA from all of the samples in the NTM positive cohort, with no significant difference (*p* = 1), except for one TW sample from a KNP elephant. The negative result for both PCRs may have been due to damaged DNA during sample storage or handling, since the same DNA sample produced a positive *M. fortuitum* result in the Hain LPA. Otherwise, the amplicon sequencing of the *rpoB* target identified all of the cultured NTM species correctly with 100% accuracy, directly from respiratory samples, unlike the sequenced *ku* amplicons that only predicted 45% of the cultured species. A significant difference (*p* < 0.02) between the *rpoB* PCR and *ku* PCR was observed for the species identification. The success of the *rpoB* PCR for culture isolate prediction is based on a study performed by Adékambi et al. (2003), where the authors focused on a 723 bp variable region, exhibiting 83.9 to 97% interspecies similarity and 0 to 1.7% interspecies divergence, to design a primer pair for both PCR amplification and sequencing of this region for the identification of rapidly growing mycobacteria [28]. Using these PCRs within the culture negative cohort, both the *rpoB* and *ku* PCRs identified the presence of *M. africanum*, *M. bovis,* and *M. orygis* DNA in three separate elephant samples. These samples were also all *ESAT-6/CFP-10*, Ultra, and Hain LPA positive, suggesting that these elephants were truly infected (Appendix A). Agreement analysis for all of the samples between tests and test combinations versus culture for genus detection and species identification revealed that the *rpoB* PCR is the ideal individual test, compared to the rest. The Hain LPA detected slightly more samples containing *Mycobacterium* genus DNA, but it differentiated the species very poorly compared to culture. Therefore, combining the *rpoB* PCR with either the *ku* PCR and/or the Hain LPA slightly increased the genus detection, but not the species identification compared to culture. All of these findings have significant consequences for species management, with regards to screening animals prior to translocation, reducing risk of spillover into humans, improving veterinary disease control to avoid further transmission between species (including livestock), and protecting susceptible endangered wildlife species against infection [40,41,42].

The limitations of this study included the use of Sanger sequencing, as opposed to deep sequencing, resulting in shallow depth coverage, limited sample size, multiple freeze–thawing of samples prior to testing, the Hain LPA specialized equipment and training requirements, and possible underestimated sensitivities and overestimated specificities when using animal samples. Lastly, the cohort only included animals from *M. bovis* and *M. tuberculosis* endemic areas. Since the animals were naturally infected, the sporadic shedding of MTBC and the lack of corroborating evidence of NTM infection could have significantly affected the findings in this study. Therefore, additional studies are needed to assess the performance of culture-independent techniques for the rapid identification of MTBC and NTM infections in livestock and wildlife. 

In conclusion, based on this pilot study, the Ultra appeared to be the optimal diagnostic test for MTBC DNA detection directly from raw antemortem respiratory specimens, and the *rpoB* PCR for *Mycobacterium* genus DNA detection and species identification through amplicon sequencing. Notably, the need for an NTM-specific test, such as the Hain LPA, could be circumvented by the combined use of the Ultra and *rpoB* PCR. 

## 4. Materials and Methods

### 4.1. Animals

Throughout 2018, 2019, and 2021, the BALF samples (*n* = 23) were opportunistically collected from free-ranging white (*n* = 21) and black (*n* = 1) rhinoceros from *M. bovis*-endemic KNP and zoo kept black (*n* = 1) rhinoceros. Trunk wash (*n* = 21) and BALF (*n* = 22) samples were also opportunistically collected from free-ranging *M. bovis*-exposed African elephants (*n* = 23) in KNP and zoo kept elephants (*n* = 3).

### 4.2. Antemortem Sample Collection and Mycobacterial Culture 

The bronchoalveolar lavage fluid was collected endoscopically, and the TW samples as previously described [5,43,44]. Briefly, 150 mL respiratory samples were collected in-field from each animal into a 500 mL sterile suction vacuum container, kept cool with ice bricks and transported to the laboratory within 4 h of collection. Thereafter, total volumes were aliquoted into three separate Corning™ Falcon™ 50 mL Conical Centrifuge Tubes (Waltham, MA, USA) per animal, concentrated by centrifugation at 2000× *g* for 30 min, pooled into a 4 mL concentrate and frozen at −20 ℃ until further processing. For optimal *Mycobacteria* spp. isolation, all antemortem respiratory samples were processed for mycobacterial culture, using the conventional Mycobacteria Growth Indicator Tubes (MGIT, Becton Dickson, Franklin Lakes, NJ, USA), in parallel with a novel decontamination and mycobacterial culture technique (TiKa) for improved sensitivity, as previously reported [8,24]. Briefly, one aliquot of sample was decontaminated using BBL MycoPrep (Becton Dickinson, Franklin Lakes, NJ, USA) and inoculated into conventional MGIT tubes containing BBL MGIT PANTA-OADC enrichment media (Becton Dickinson, Franklin Lakes, NJ, USA), as described by the manufacturers. A second aliquot of the sample was decontaminated using TiKa-KiC (TiKa Diagnostics, London, UK) and inoculated into MGIT-TiKa tubes containing PANTA-OADC enrichment media and TiKa supplement B reagent (TiKa Diagnostics, London, UK). All of the MGITs (conventional and TiKa) were transferred to the BACTEC MGIT 960 mycobacterial detection system (Becton Dickinson). All of the liquid cultures with detected bacterial growth were sub-cultured onto blood agar plates to exclude contaminants and were subjected to Ziehl–Neelsen (ZN) acid fast staining [45]. All of the MGITs (conventional and TiKa) with positive bacterial growth underwent a further *Mycobacteria* spp. genetic speciation using the region of difference PCR, *16S* rRNA, *rpoB* and *hsp65* PCRs, and Sanger sequencing (Illumina Inc., San Diego, CA, USA), as previously described [28,29,46]. The animals with samples confirmed to contain either MTBC and/or NTM by culture and genetic speciation were defined as “MTBC positive” or “NTM positive”. Furthermore, the presence or absence of immunogenic proteins in all of the bacterial growth-positive MGIT tubes were confirmed by PCR amplification and subsequent amplicon sequencing of *ESAT-6* and *CFP-10* genes, as previously described [47]. 

### 4.3. Nucleic Acid Amplification Tests (NAAT) for Mycobacteria spp. Detection and Differentiation

The GeneXpert MTB/RIF Ultra assay (Cepheid, Sunnyvale, CA, USA) was performed on the raw BALF and TW samples for the detection of MTBC DNA, as previously described [5]. Briefly, 700 µL of samples were treated to chemical lysis, as prescribed by the manufacturer, and 2 mL of the solution aliquoted into the GeneXpert MTBC/RIF Ultra cartridge sample chamber. The result outputs were as follows: (1) “MTB not detected”; (2) “MTB trace detected”; and (3) “MTB detected high/medium/low/very low” [5]. 

Prior to performing the remaining NAATs, the total DNA was first extracted from all of the raw BALF and TW samples using the QIAGEN DNeasy^®^ Blood and Tissue kit (Qiagen, Hilden, Germany), as previously described [48]. The DNA was used in the Genotype CM*direct* version 1.0, line probe assay (Hain Lifescience, Germany) to determine the presence of the *Mycobacterium* genus, MTBC (without differentiation) and the subsequent detection and differentiation of more than 20 clinically relevant NTM species, according to manufacturer’s instructions. Briefly, the multiplex PCRs were performed using biotinylated primers and proprietary amplification mixes in a MiniAmp^TM^ Thermal Cycler (ThermoFisher Scientific, Waltham, MA, USA) with a ramp rate of ≤2.2 °C/s. Thereafter, the amplicons were reverse hybridized in an automated GT-Blot 48 hybridization washer (Bruker, Billerica, MA, USA) onto precoated membrane strips and the hybridization results interpreted according to a template provided within the kit. 

The extracted DNA was also subjected to conventional PCR using *Mycobacterium*-specific primers for two highly conserved genetic regions, the *ku* (600 bp) and *rpoB* (740 bp) genes [26,27,28,29]. The amplicons were speciated by Sanger sequencing (Illumina, Inc.), as previously described [27,28,29]. Briefly, for both targets, a total reaction volume of 25 µL was prepared that consisted of 12.5 µL OneTaq Hot Start 2x master mix (New England Biolabs, Ipswich, MA, USA), 0.8 µM of each respective forward and reverse primer (Integrated DNA Technologies, Coralville, IA, USA), and 6.5 µL nuclease-free water and 5 µL DNA template. The positive controls which included 5 µL 30 ng/µL extracted H37Rv *M. tuberculosis* and *M. bovis* DNA, as well as no template controls, were used in each PCR. Using a Veriti^TM^ 96-well Thermal Cycler (Applied Biosystems, Waltham, MA, USA), the cycling conditions were 95 °C for 15 min, followed by 45 cycles at 95 °C for 1 min, 64 ℃ for 1 min, and 72 °C for 1 min, and a final elongation at 72 °C for 5 min. The amplicon presence, size, and intensity were confirmed by 1.5% agarose gel electrophoresis, followed by gel imaging using the ChemiDoc M.D. Universal Hood III Gel Documentation System (Bio-Rad, Hercules, CA, USA). All of the amplicons were sequenced by Sanger sequencing (Illumina, Inc., San Diego, CA, USA) through the Central Analytical Facility (CAF) at Stellenbosch University, SA. The sequence contigs were generated using Sequencher 5.1. software (Gene Codes, Ann Arbor, MI, USA) and blasted on NCBI’s Basic Local Alignment Search Tool for nucleotide (BLASTn) [30].

### 4.4. Data Analysis

All of the positive and negative NAAT and mycobacterial culture results are reported as proportions of the total number of animals and respiratory samples tested. For all of the known *Mycobacteria* spp. (MTBC and NTM) culture-positive specimens, each relevant NAAT and combination of NAATs with similar target species, the test-positive proportions were compared using a two-tailed z-test, where the z-scores and *p*-values were calculated (https://www.socscistatistics.com/tests/ztest/default2.aspx (accessed on 25 May 2022)). The *p*-values were considered statistically significant if *p* < 0.05. Thereafter, agreement analysis was performed between the relevant individual NAATs and test combinations for genus detection and differentiation, MTBC, and NTM detection and differentiation by calculating Cohen’s kappa coefficient (κ) using the online agreement calculator webtool (https://graphpad.com/quickcalcs/kappa1/ (accessed on 25 May 2022)). All of the NAATs and culture results for each sample are provided in Appendix A.

## Figures and Tables

**Figure 1 pathogens-11-00709-f001:**
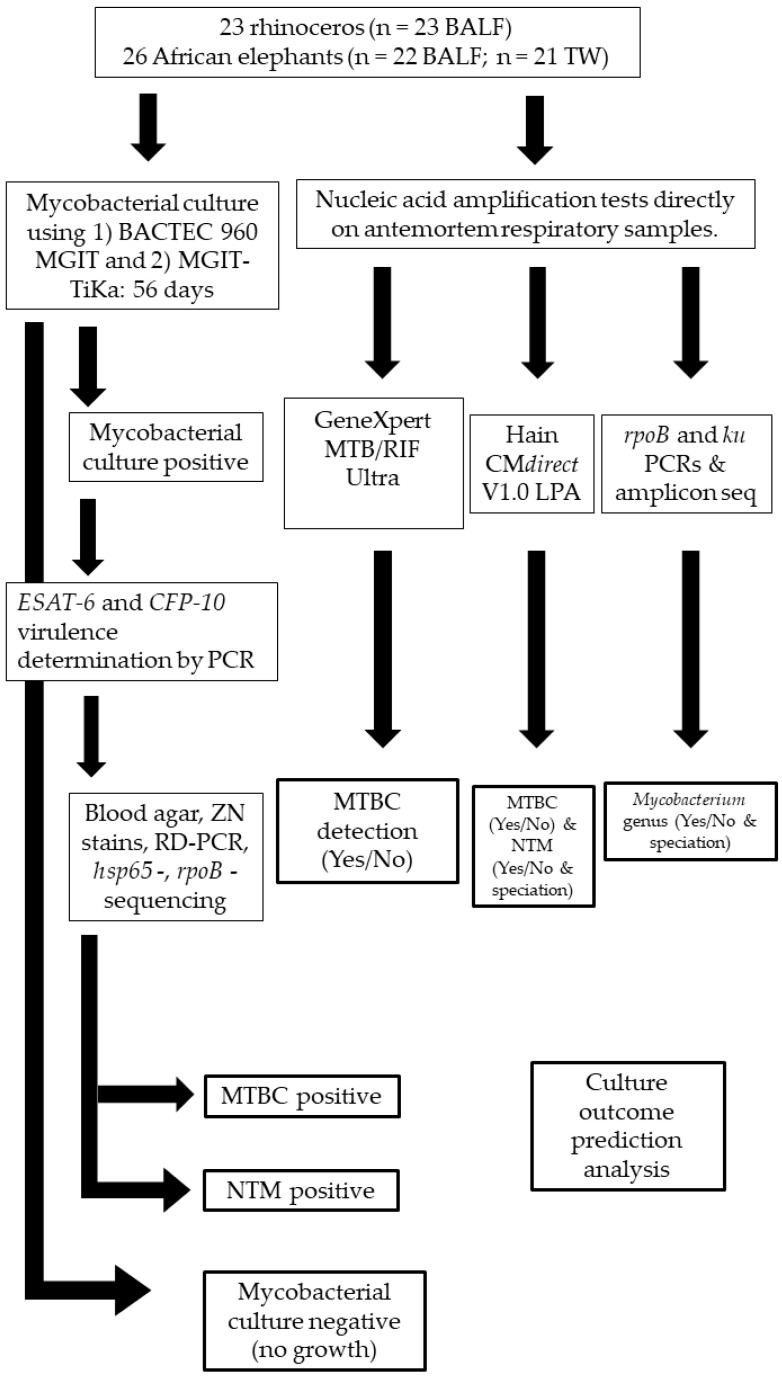
Study method flow chart for African elephants (*n* = 26) and rhinoceros (*n* = 23) respiratory sample processing and PCR testing for mycobacterial identification. TW: Trunk wash; BALF: Bronchioalveolar lavage fluid; RD: Region of difference; MTBC: *Mycobacterium tuberculosis* complex; NTM: Non-tuberculous mycobacteria.

**Table 1 pathogens-11-00709-t001:** Mycobacterial isolates identified by culture and virulence determination of *n* = 20 respiratory samples from 13 African elephants and 2 white rhinoceros, including the combinational use of four nucleic acid amplification tests directly on respiratory specimens for the prediction of culture outcome.

				Mycobacterial Culture Result from Antemortem Respiratory Specimens		Nucleic Acid Amplification Test Results from Antemortem Respiratory Specimens			
*Mycobacterial* spp. Present	Species (#Animals)	Sample Type (Animal ID)	Location (South Africa)	Combined MGIT and MGIT-TiKa Result	ESAT-6 and CFP-10 Virulence Determination	GeneXpert MTB/RIF Ultra Result	Hain CM*direct* V1.0 LPA Result	*rpoB* PCR Result	*ku* PCR Result
Confirmed MTBC-positive	African elephants *n* = 5	BALF (18/85)	Zoo	*M. tuberculosis*	Positive	MTB DETECTED Medium; RIF resistance indeterminate	MTBC	*M. tuberculosis*	*M. tuberculosis*
		BALF (18/177)	Zoo	*M. tuberculosis*	Positive	MTB TRACE DETECTED	Mixed NTM	*M. tuberculosis*	*M. tuberculosis*
		BALF (18/177)	Zoo	*M. africanum*	Positive	MTB TRACE DETECTED	*M. avium* and/or *M. interjectum*	*M. africanum*	*M. africanum*
		TW (18/177)	Zoo	*M. elephantis strain*	Positive	MTB NOT DETECTED	*M. fortuitum* group	*M. elephantis*	*M. fortuitum* group
		BALF (18/527)	KNP	*M. stomatepiae*	Negative	MTB NOT DETECTED	MTBC and/or *M. fortuitum* group	*M. stomatepiae*	mixed NTMs-*M. smegmatis*
		TW (18/527)	KNP	*M. bovis*	Positive	MTB TRACE DETECTED	Mixed NTM	*M. bovis*	*M. bovis*
		TW (18/533)	KNP	*M. bovis*	Positive	MTB TRACE DETECTED	MTBC and/or mixed NTM infection	*M. bovis*	*M. bovis*
		TW (18/538)	KNP	*M. bovis*	Positive	MTB TRACE DETECTED	*M. fortuitum* group	*M. bovis*	*M. bovis*
	White rhinoceros *n* = 1	BALF (19/46)	KNP	*M. bovis*	Positive	MTB TRACE DETECTED	MTBC and/or mixed NTM infection	*M. bovis*	*M. bovis*
Confirmed NTM-positive	African elephants *n* = 8	TW (18/173)	KNP	*M. avium complex strain*	Positive	MTB TRACE DETECTED	MTBC and/or mixed NTM infection	*M. avium* complex	*Mycobacteria* spp.
		BALF (18/176)	Zoo	*M. foliorum*	Positive	MTB NOT DETECTED	*M. fortuitum* group	*M. fortuitum* group	*M. fortuitum* group
		BALF (19/460)	KNP	*M. mantenii*	Positive	MTB TRACE DETECTED	MTBC and/or mixed NTM infection	*M. mantenii*	*M. mantenii*
		TW (19/460)	KNP	*M. abscessus strain*	Positive	MTB NOT DETECTED	*M. fortuitum* group and/or *M. abscessus*	*M. abscessus*	*M. abscessus*
		BALF (18/530)	KNP	*M. interjectum strain*	Positive	MTB NOT DETECTED	Mixed NTM	*M. interjectum strain*	mixed NTMs-*M. avium* complex
		TW (18/530)	KNP	*M. avium strain*	Negative	MTB NOT DETECTED	Mixed NTM	*M. avium* complex	*M. elephantis* strain
		TW (18/532)	KNP	*M. mageritense strain*	Negative	MTB NOT DETECTED	*M. fortuitum* group	Negative	Negative
		TW (18/534)	KNP	*M. intracellulare*	Positive	MTB TRACE DETECTED	MTBC and/or mixed NTM infection	*M. avium* complex	*M. intracellulare*
		TW (18/539)	KNP	*M. intracellulare*	Negative	MTB NOT DETECTED	*M. fortuitum* group	*M. intracellulare*	mixed NTMs-*M. avium* complex
		TW (21/496)	KNP	*M. fortuitum strain*	Positive	MTB NOT DETECTED	*M. fortuitum* group	*M. fortuitum*	*M. elephantis* strain
	White rhinoceros *n* = 1	BALF (18/31)	KNP	*M. scrofulaceum strain*	Positive	MTB NOT DETECTED	Mixed NTM	*M. avium subsp. Paratuberculosis*	mixed NTMs-*M. avium subsp. Paratuberculosis*

BALF: Bronchoalveolar lavage fluid; TW: Trunk wash; KNP: Kruger National Park; MGIT: Measurable growth in the *Mycobacterium* growth indicator tube (MGIT) detected by the BACTEC MGIT 960 mycobacterial detection system and strain typing (Warren et al., 2016); TiKa: modified MGIT system by alternative use of cationic D-enantiomer peptides for sample decontamination and the additional use of Supplement B during tube inoculation (Goosen et al., 2022); *ESAT-6/CFP-10*: PCR amplification of both virulence genetic targets from all culture; GeneXpert MTB/RIF Ultra: MTBC DNA detection through probe-based qPCR that simultaneously targets insertion elements IS*6110* and IS*1081* and, if positive, targets *rpoB* for drug resistance determination; Hain CM*direct* V1.0 line probe assay is a test system for the detection of *M. tuberculosis* complex and differentiation of more than 20 clinically relevant NTM directly from patient specimens; *rpoB* PCR uses *Mycobacterium* genus specific primers for the identification of genus DNA and subsequent speciation through amplicon sequencing; *Ku* PCR uses *Mycobacterium* genus specific primers for the identification of genus DNA and subsequent speciation through amplicon sequencing.

**Table 2 pathogens-11-00709-t002:** The kappa (κ), 95% confidence interval, standard error (SE) of agreement between four tests and culture for antemortem *Mycobacterium* genus DNA detection from respiratory samples from African elephants (*n* = 26) and rhinoceros (*n* = 23).

Test and Combinations	*rpoB* PCR	*ku* PCR	Hain LPA	Culture	*rpoB/ku*	*rpoB*/Hain LPA	*ku*/Hain LPA	*rpoB*/*ku*/Hain LPA
*rpoB* PCR	1							
*ku* PCR	0.91 (0.81–1.00, 0.05)	1						
Hain LPA	0.94 (0.85–1.00, 0.04)	0.88 (0.76–0.99, 0.06)	1					
Culture	0.47 (0.29–0.66, 0.09)	0.50 (0.31–0.68, 0.09)	0.32 (0.16–0.48, 0.09)	1				
*rpoB*/*ku* PCRs	0.97 (0.91–1.00, 0.03)	0.94 (0.86–1.00, 0.04)	0.97 (0.91–1.00, 0.03)	0.45 (0.27–0.63, 0.09)	1			
*rpoB*/Hain LPA	0.94 (0.85–1.00, 0.04)	0.94 (0.85–1.00, 0.04)	1.00 (0.95–1.00, 0.03)	0.46 (0.29–0.64, 0.09)	0.94 (0.85–1.00, 0.04)	1		
*ku*/Hain LPA	0.94 (0.85–1.00, 0.04)	0.94 (0.85–1.00, 0.04)	1.00 (0.95–1.00, 0.03)	0.46 (0.29–0.64, 0.09)	0.94 (0.85–1.00, 0.04)	1.00 (0.95–1.00, 0.03)	1	
*rpoB*/*Ku*/Hain LPA	0.94 (0.85–1.00, 0.04)	0.94 (0.85–1.00, 0.04)	1.00 (0.95–1.00, 0.03)	0.46, (0.29–0.64, 0.09)	0.94 (0.85–1.00, 0.04)	1.00 (0.95–1.00, 0.03)	1.00 (0.95–1.00, 0.03)	1

*rpoB* PCR uses *Mycobacterium* genus specific primers for the identification of genus DNA and subsequent speciation through amplicon sequencing; *Ku* PCR uses *Mycobacterium* genus specific primers for the identification of genus DNA and subsequent speciation through amplicon sequencing; Hain CM*direct* V1.0 line probe assay is a test system for the detection of *M. tuberculosis* complex and differentiation of more than 20 clinically relevant NTM directly from patient specimens; Mycobacterial culture detection based on the combined isolation of *Mycobacteria* spp. through conventional BACTEC 960 MGIT system and MGIT-TiKa, followed by appropriate genetic speciation of all isolates (Warren et al., 2016). Landis Kappa scale: Kappa < 0: No agreement; Kappa between 0.00 and 0.20: Slight agreement; Kappa between 0.21 and 0.40: Fair agreement; Kappa between 0.41 and 0.60: Moderate agreement; Kappa between 0.61 and 0.80: Substantial agreement; Kappa between 0.81 and 1.00: Almost perfect agreement.

## Data Availability

Data are available in the Appendix A.

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
