# Peer review of "Culture-Independent PCR Detection and Differentiation of Mycobacteria spp. in Antemortem Respiratory Samples from African Elephants (Loxodonta Africana) and Rhinoceros (Ceratotherium Simum, Diceros Bicornis) in South Africa"

_pathogens, 2022, doi:10.3390/pathogens11060709_

Round 1
Reviewer 1 Report
In general, the article shows the importance of MTBC infections and the problem of the presence of NTM and the difficulty of a differential diagnosis between this group of bacteria; the writing is well argued with a logical sequence and the object of study is established.
The methodology is not new, however it is current and according to its results and that of previous studies gives certainty of the results shown.
It is important to comment that during the introduction and discussion, the sensitivity of the bacteriological isolation test is indicated several times; it is documented in several articles what could be the minimum concentration where it can be recovered for isolation, however there are no citations to support this; They also indicate that the sample's bacterial load may also be reduced during sampling, sample handling and storage, but they do not indicate how they were preserved or the time that passed until processing. Maybe you could describe something about this.
45% of the cited references are within the last 5 years. Perhaps you could increase the number of more recent journals.
The results table schemes appropriate and do they properly show the data. The statistical analysis is adequate and accessible to follow. The number of lines is within the average for similar works. In Figure 1, where the Study method flow is shown, it does not indicate more than what is already described in the text, so it may not be necessary.
Author Response
We truly thank the reviewer taking the time to carefully review our manuscript. All suggested changes have now been considered by all the authors and addressed accordingly. We thank the reviewer for truly contributing and improving our study. Please find specific responses below.
Reviewer #1
In general, the article shows the importance of MTBC infections and the problem of the presence of NTM and the difficulty of a differential diagnosis between this group of bacteria; the writing is well argued with a logical sequence and the object of study is established.
The methodology is not new, however it is current and according to its results and that of previous studies gives certainty of the results shown.
It is important to comment that during the introduction and discussion, the sensitivity of the bacteriological isolation test is indicated several times; it is documented in several articles what could be the minimum concentration where it can be recovered for isolation, however there are no citations to support this;
Response:
We have now included specific statements in both the introduction and discussion indicating the lower limit of detection of mycobacterial culture when used specifically on rhinoceros and elephant respiratory specimens. This is reference is relevant to the current study in rhinoceros and elephants.
Introduction:
“Previously, using elephant and rhinoceros’ respiratory samples, mycobacterial culture has shown to have a lower limit of detection of 1000 colony forming units (CFU) for culturing M. bovis and 100 CFU for M. tuberculosis [23,24].”
Discussion:
“However, one limitation was that all these animals were from MTBC endemic areas and M. tuberculosis and M. bovis may have been present in small amounts below the detection threshold of 100 CFU for M. tuberculosis culture and 1000 CFU for M. bovis, but detectable by PCR [5,24].”
Reviewer #1:
They also indicate that the sample's bacterial load may also be reduced during sampling, sample handling and storage, but they do not indicate how they were preserved or the time that passed until processing. Maybe you could describe something about this.
Response:
Under material and methods, we now describe collection as follows:” Bronchial alveolar lavage fluid was collected endoscopically, and TW samples as previously described [5,43,44]. Briefly, 150 ml respiratory samples were collected in-field from each animal into a 500 ml sterile suction vacuum container, kept cool with ice bricks and transported to the laboratory within 4 hours of collection. Thereafter, total volumes were aliquoted into three separate Corning™ Falcon™ 50 mL Conical Centrifuge Tubes (Waltham, Massachusetts, USA) per animal, concentrated by centrifugation at 2 000 x g for 30 min, pooled into a 4 ml concentrate and frozen at -20 ℃ until further processing.”
Reviewer #1:
45% of the cited references are within the last 5 years. Perhaps you could increase the number of more recent journals.
Response:
We appreciate this suggestion. All authors have carefully considered which older reference may be replaced and decided the replace the following reference numbers: 12, 13, 14, 15, 18, 19 and 36. Some older references, especially those regarding rpoB were kept due to their importance to the use of rpoB in this study directly on antemortem samples from wildlife.
Reviewer #1:
The results table schemes appropriate and do they properly show the data. The statistical analysis is adequate and accessible to follow. The number of lines is within the average for similar works. In Figure 1, where the Study method flow is shown, it does not indicate more than what is already described in the text, so it may not be necessary.
Response:
We agree with the reviewer that Fig. 1 is a duplicate of what is described in the text. We do not mind removing it, if the editorial team support its removal.
Reviewer 2 Report
The manuscript describes a pilot study investigating and comparing several techniques to detect and identify species of Mycobacteria, both Mycobacterium tuberculosis complex members (e.g. M. africanum, M. bovis) and non-tuberculosis mycobacteria (e.g. M. avium, M. abscessus) in bronchoalveolar lavage fluid from elephants and rhinoceros and trunk washes from elephants. The manuscript is well written, but sometimes difficult to follow, as there are many results discussed involving multiple spp. of mycobacteria.
My main issue is with Table 1. Firstly, it needs to be on a single page or the headings need to be repeated. The table needs to be laid out in a more reader-friendly manner. Using shading or lines to demarcate between the different samples would be useful. The first two columns needs to be rethought.
Line 19 – GeneXpert MTB and Hain CMdirect V1.0 are given slightly different names in the rest of the manuscript, except the figure. Please be consistent throughout.
Line 158 – make it clear that this may account for the positive result.
Line 302 – remove ‘like BALF and TW’ as unnecessary.
Author Response
We truly thank the reviewer taking the time to carefully review our manuscript. All suggested changes have now been considered by all the authors and addressed accordingly. We thank the reviewer for truly contributing and improving our study. Please find specific responses below.
Reviewer #2
The manuscript describes a pilot study investigating and comparing several techniques to detect and identify species of Mycobacteria, both Mycobacterium tuberculosis complex members (e.g. M. africanum, M. bovis) and non-tuberculosis mycobacteria (e.g. M. avium, M. abscessus) in bronchoalveolar lavage fluid from elephants and rhinoceros and trunk washes from elephants. The manuscript is well written, but sometimes difficult to follow, as there are many results discussed involving multiple spp. of mycobacteria.
My main issue is with Table 1. Firstly, it needs to be on a single page or the headings need to be repeated. The table needs to be laid out in a more reader-friendly manner. Using shading or lines to demarcate between the different samples would be useful. The first two columns needs to be rethought.
Response:
We thank the reviewer for highlighting this concern. Table 1 now fits on a single page. We have also re-labelled column 1 as “Mycobacterial species present” and have separated all sample results based on confirmed MTBC positive (elephants and rhinoceros) and NTM positive animals (elephants and rhinoceros) to improve describing test responses within each cohort. This change will allow the reader to discern if there was detection of MTBC or NTMs from different samples collected in the same individual animal.
Reviewer #2:
Line 19 – GeneXpert MTB and Hain CMdirect V1.0 are given slightly different names in the rest of the manuscript, except the figure. Please be consistent throughout.
Response:
We thank the reviewer for picking this up and have now made sure that their official names GeneXpert MTB/RIF Ultra and Hain CMdirect V1.0 LPA are used and defined appropriately at the start of the manuscript. Thereafter, both are defined as Ultra and Hain LPA for the rest of the manuscript, respectively.
Reviewer #2:
Line 158 – make it clear that this may account for the positive result.
Response: This is now indicated clearly in the Discussion.
Reviewer #2:
Line 302 – remove ‘like BALF and TW’ as unnecessary.
Response:
This has now been removed.
Reviewer 3 Report
The authors explored the different diagnostic capabilities of some of the more used MTBC diagnostic methods, with the addition of a more recent design method using two target genes present in the Mycobacterium genus genome. This study is important in the field of wildlife animal TB diagnostic and control.
However, this reviewer has some comments/suggestions regarding this manuscript.
Abstract
Line 21 - please add information regarding the type of respiratory samples (BALF and TW, together with the number of samples of each type).
Line 23 - please use the full test name when referring to the GeneXpert MTB/RIF Ultra.
2. Results
Legend of figure 1 -NTM - non-tuberculous mycobacteria; please rectify.
The lack of verticle lines in Table 1 makes it difficult to clearly perceive which samples are MTBC and NTM, and which are from rhinos or elephants. Please add these verticle lines.
Legend of table 1 and table 2 should show italicized species names; please rectify this.
The supplementary table does not state animal ID or location (zoo or KNP), so the replicate samples and association between BALF and TW can not be made by the reader; please add this information to the supplementary table.
Line 225 - please italicize genes names and Mycobacterium
3. Discussion
Line 376 - please italicize Mycobacterium
Line 381 - please italicize rpoB
Line 396 - please italicize Mycobacterium
Line 413 - please italicize Mycobacterium
Line 399 to 400 - Please elaborate more on the specific consequences of these findings to the different mentioned fields.
4. Materials and Methods
Authors should standardize the way they indicate the number of samples; in some cases, the number of samples is shown before the host (e.g.) and sometimes after.
Lines 420 to 422 - the number of BALF or TW of African elephants is not explicitly stated, only the number of sampled animals is clearly stated; please add this information.
The abbreviation need also to be standardized since the first time in the section authors use the acronym BALF, but in the next paragraph use the complete terminology "Bronchial alveolar lavage fluid".
Line 237 to 447 - In agreement with all other methods used, in this section, the authors should also briefly explain the used PCR protocols.
Line 472 to 474 + 480 - replace the comma on numbers with a dot.
This reviewer believes that authors should deposit all obtained sequences in NCBI and make the accession number available on the manuscript (e.g. adding it to the supplementary table).
Author Response
We truly thank the reviewer taking the time to carefully review our manuscript. All suggested changes have now been considered by all the authors and addressed accordingly. We thank the reviewer for truly contributing and improving our study. Please find specific responses below.
Reviewer #3:
The authors explored the different diagnostic capabilities of some of the more used MTBC diagnostic methods, with the addition of a more recent design method using two target genes present in the Mycobacterium genus genome. This study is important in the field of wildlife animal TB diagnostic and control.
However, this reviewer has some comments/suggestions regarding this manuscript.
Abstract
Line 21 - please add information regarding the type of respiratory samples (BALF and TW, together with the number of samples of each type).
Response:
The number of specific samples have now been added under Materials and Methods, 4.1. Animals and type and number of samples in the abstract.
Reviewer #3:
Line 23 - please use the full test name when referring to the GeneXpert MTB/RIF Ultra.
Response:
This has also been suggested by n a previous reviewer and have now been amended throughout the manuscript.
Reviewer #3:
- Results
Legend of figure 1 -NTM - non-tuberculous mycobacteria; please rectify.
Response:
This has now been rectified.
Reviewer #3:
The lack of verticle lines in Table 1 makes it difficult to clearly perceive which samples are MTBC and NTM, and which are from rhinos or elephants. Please add these verticle lines.
Response:
We agree with this. We have now added horizontal lines and believe this does improve differentiating between MTBC and NTM positive animals and their respective samples.
Reviewer #3:
Legend of table 1 and table 2 should show italicized species names; please rectify this.
Response:
Thank you to the reviewer for picking this up. We have now amended accordingly.
Reviewer 3:
The supplementary table does not state animal ID or location (zoo or KNP), so the replicate samples and association between BALF and TW cannot be made by the reader; please add this information to the supplementary table.
Response:
This has now been included.
Reviewer #3:
Line 225 - please italicize genes names and Mycobacterium
Response:
This has now been done throughout the manuscript.
Reviewer #3:
- Discussion
Line 376 - please italicize Mycobacterium
Line 381 - please italicize rpoB
Line 396 - please italicize Mycobacterium
Line 413 - please italicize Mycobacterium
Response:
This has now been done throughout the manuscript.
Reviewer #3:
Line 399 to 400 - Please elaborate more on the specific consequences of these findings to the different mentioned fields.
Response:
This now reads: “All these findings have significant consequences for species management with regards to screening animals prior to translocation, reducing risk of spillover into humans, improving veterinary disease control to avoid further transmission between species (including livestock), and protecting susceptible endangered wildlife species against infection [32,41,42].
Reviewer #3:
- Materials and Methods
Authors should standardize the way they indicate the number of samples; in some cases, the number of samples is shown before the host (e.g.) and sometimes after.
Response:
We thank the reviewer for identifying this, we have now changed this accordingly throughout the manuscript (including Abstract).
Reviewer #3:
Lines 420 to 422 - the number of BALF or TW of African elephants is not explicitly stated, only the number of sampled animals is clearly stated; please add this information.
Response:
We have changed this now based on a previous comment requesting this in the abstract as well.
Reviewer #3:
The abbreviation need also to be standardized since the first time in the section authors use the acronym BALF, but in the next paragraph use the complete terminology "Bronchial alveolar lavage fluid".
Response:
This has now been addressed.
Reviewer #3:
Line 237 to 447 - In agreement with all other methods used, in this section, the authors should also briefly explain the used PCR protocols.
Response:
All PCR protocols are described, and their developers appropriately referenced under Materials and Methods, section 4.3. Nucleic acid amplification tests (NAAT) for Mycobacteria spp. detection and differentiation. These include the GeneXpert MTB/RIF Ultra, Hain CMdirect V1.0 LPA, rpoB PCR and ku PCR with reaction preparations and cycling conditions. Additional information is also available in the referenced papers describing the development of all these tests.
For the ku and rpoB PCRs: “Briefly, for both targets, a total reaction volume of 25 µl was prepared that consisted of 12.5 µl OneTaq Hot Start 2x master mix (New England Biolabs, Ipswich, MA, USA), 0.8 µM of each respective forward and reverse primer (Integrated DNA Technologies, Coralville, IA, USA), and 6.5 µl nuclease-free water and 5 µL DNA template. Positive controls which included 5 µL 30ng/µl extracted H37Rv M. tuberculosis and M. bovis DNA, as well as no template controls were used in each PCR. Using a VeritiTM 96-well Thermal Cycler (Applied Biosystems, Waltham, MA, USA), cycling conditions were 95℃ for 15 minutes, followed by 45 cycles at 95℃ for 1 minute, 64℃ for 1 minute and 72℃ for 1 minute, and final elongation at 72℃ for 5 minutes. Amplicon presence, size and intensity were confirmed by 1.5% agarose gel electrophoresis, followed by gel imaging using the ChemiDoc M.D. Universal Hood III Gel Documentation System (Bio-Rad, Hercules, CA, USA). All amplicons were sequenced by Sanger sequencing (Illumina, Inc.) through the Central Analytical Facility (CAF) at Stellenbosch University, SA. Sequence contigs were generated using Sequencher 5.1. software (Gene Codes, Ann Arbor, MI, USA) and blasted on NCBI’s Basic Local Alignment Search Tool for nucleotide (BLASTn)[29].”
Reviewer #3:
Line 472 to 474 + 480 - replace the comma on numbers with a dot.
Response:
This has now been done.
Reviewer #3:
This reviewer believes that authors should deposit all obtained sequences in NCBI and make the accession number available on the manuscript (e.g., adding it to the supplementary table).
Response:
In this study the use of already developed, described, and published PCR tests (and primers) were evaluated directly on raw respiratory samples from elephants and rhinoceros with known mycobacterial culture results. This includes the use of published primers and the subsequent BLASTn of obtained amplicon sequences against existing sequences on NCBI’s gene database. Without the existing sequences on NBCI’s database, species identification using these two PCRs would not have been possible. Therefore, submitting our sequences will only be a duplication of what is already available online. No new information will be added. The novel contribution of this study is the culture independent use of these existing published tools directly on antemortem wildlife samples to predict culture outcome.